# Soil Electrical Conductivity and Satellite-Derived Vegetation Indices for Evaluation of Phosphorus, Potassium and Magnesium Content, pH, and Delineation of Within-Field Management Zones

Piotr Mazur [1], Dariusz Gozdowski [2] and Elżbieta Wójcik-Gront [2,*]

[1]  Agrotechnology, Jagiellonów 4, 73-150 Łobez, Poland; pmazur@agrotechnology.pl
[2]  Department of Biometry, Institute of Agriculture, Warsaw University of Life Sciences, Nowoursynowska 159, 02-776 Warsaw, Poland; dariusz_gozdowski@sggw.edu.pl
*  Correspondence: elzbieta_wojcik_gront@sggw.edu.pl; Tel.: +48-225-932-645

**Abstract:** The optimization of soil sampling is very important in precision agriculture. The main aim of this study was to evaluate the relationships between selected spectral indices (NDWI—normalized difference water index and NDVI—normalized difference vegetation index) and apparent soil electrical conductivity (EC) with soil nutrient content (phosphorus, potassium, and magnesium) and pH. Moreover, the usefulness of these variables for the delineation of within-field management zones was assessed. The study was conducted in 2021 in central Poland at three maize fields with a total area approximately 100 ha. The analyses were performed based on 47 management zones, which were used for soil sampling. Significant positive correlations were observed between the NDVI for the bare soil and all the studied nutrient contents in the soil and pH. A very strong positive correlation was observed between the soil EC and the potassium content and a moderate correlation was found with the magnesium content. A multiple-regression analysis proved that the soil nutrient content, especially potassium and phosphorus, was strongly related to the EC and NDVI. The novelty of this study is that it proves the relationships between soil and the crop attributes, EC and NDVI, which can be measured at field scale relatively simply, and the crucial soil nutrients, phosphorus and potassium. This allows the results to be used for optimized variable-rate fertilization.

**Keywords:** precision agriculture; variable rate fertilization; management zones; remote sensing; soil electrical conductivity

## 1. Introduction

Precision agriculture is a crop management technology whose main aim is to optimize crop fertilization based on site-specific crop management, using information on the spatial variability of soils' physical-chemical properties and crop growth [1]. Variable-rate fertilization can be applied based on within-field management zones, i.e., homogenous areas, according to the main agronomic properties of soil, such as the soil texture, nutrient contents, and organic matter [2]. The delineation of management zones can be performed not only with the use of information on soil variability, but also information on the spatial variability of crops, in which such variables as the vegetation indices characterizing crop growth or crop yield are related to soil conditions [3,4]. The most advanced methods for the delineation of management zones use a combination of different types of multivariate data [5–7]. When a large number of variables is used for the delineation of management zones, principal component analysis or weighted multivariate spatial clustering can be applied to obtain the optimal selection of homogenous areas within crop fields [8–10]. The multivariate algorithms used for the delineation of management zones demand many variables and advanced analysis, which, in practice can be difficult to conduct. Therefore,

simplified approaches based on univariate analyses are more common in agricultural practice. In this approach to the delineation of management zones, one variable is used, and it is strongly correlated with major agronomic soil properties and yielding potential. A common variable used for such purposes is the apparent electrical conductivity (EC) of soil [11,12]. EC can be measured at field scale using various types of equipment that allow for the quick and reliable mapping of EC variability [13,14]. The soil EC is usually correlated with soil texture (i.e., the percentage of sand, silt, and clay) which is a very important physical soil property that strongly affects yield potential [15,16]. EC can be used as an indirect indicator of soil nutrient content and other important soil physicochemical properties, such as soil pH and soil organic matter content [17]. The disadvantage of the soil EC is the necessity of using special agricultural machinery for soil scanning, the selection of appropriate conditions, i.e., moderate moisture in periods without vegetation cover. The other variables that have been commonly used for the delineation of management zones in recent years are spectral indices, which are based on remote sensing satellite-derived or UAV (unmanned aerial vehicle) data [18,19]. Some common spectral indices used to select management zones are the NDVI (normalized difference vegetation index) [18] and the NDWI (normalized difference water index) [20] or other spectral indices based on visible and infrared light.

In a more complex approach, two or more soil or crop attributes can be used for the delineation of management zones for site-specific crop management. These variables include the spectral index (e.g., NDVI) and the soil EC, since both are relatively easy to acquire [21,22]. This method allows the selection of zones that are differentiated according to different soil properties, such as the nutrient content, the soil texture (content of sand and clay), and other physico-chemical soil properties that are important for crop management. The vegetation indices used for these purposes can be satellite-derived or acquired using unmanned aerial vehicles (UAV) [21,23]. Combining EC maps and NDVI maps or other variables for the delineation of management zones can be performed using simple criteria (e.g., values below/above the mean or median) or based on more advanced multivariate methods (e.g., principal component analysis–PCA) [23,24]. In this approach, one or two principal components (PC1 and PC2) representing multivariate data are used for the selection of management zones. The soil EC (measured by electromagnetic induction—EMI) can be used, together with NDVI, for the delineation of management zones processed by the fuzzy *c*-means (FCM) clustering method [25]. This approach makes it possible to delineate the within-field management zones that differ in terms of the most important soil properties for crop management (i.e., soil texture, soil moisture deficit, yield, and nitrate residue). The delineation of management zones can be performed using the NDVI either alone or in combination with the NDWI, as well as with the EC. This approach was applied in the study by Serrano et al. [26], in which pasture management zones with different yield potential were delineated for optimized variable-rate fertilization.

In this study, the relationships between satellite-derived spectral indices (NDVI and NDWI) and the soil electrical conductivity and the content of soil nutrients, such as potassium, phosphorus, magnesium, and soil pH were evaluated. The main aim was to determine which of these variables (spectral indices or EC) was more appropriate for the delineation of management zones that can be used for soil sampling in precision agriculture. Moreover, the simultaneous effect of two variables (EC and spectral indices) on soil nutrients was evaluated for the better prediction of soil conditions, which can be used for the optimization of variable-rate fertilization.

## 2. Materials and Methods

### 2.1. Study Area

This study was conducted in 2021, in central Poland (51°10′51″ N 21°09′04″ E), at three crop fields with maize. The areas of the fields were 23.4 ha, 25.4 ha, and 52.4 ha. Within the fields, 47 management zones for soil sampling were delineated. The delineation was based on the soil EC to obtain a homogenous soil according to the soil texture within

each zone. The mean area of an individual management zone was 1.94 ha, with a standard deviation of 0.79 ha and a range from 0.53 to 3.69 ha. Such individual zones are usually used in agronomic practice for fertilization with a variable rate using precision-agriculture technology. The zones were used for soil sampling to evaluate nutrient content (potassium, phosphorus, magnesium) and pH. Standard crop rotation with main crops, such as cereals and rapeseed, and crop fertilization (NPK–nitrogen, phosphorus and potassium) was applied in the studied fields during the years prior to the year of the research. Fertilization was adjusted to crop requirements at expected yields. For the year of the study, the rates of mineral fertilizers were as follows: 115 kg N, 63 kg $P_2O_5$, and 63 kg $K_2O$ per hectare. The dates of maize sowing were from 28th of April 2021 to 1st of May 2021. The harvest of maize at full maturity was conducted at the end of October.

*2.2. Soil Sampling and Mapping of Soil Properties*

The proximal sensing methods were used to map the within-field soil variability. The soil of the studied fields was examined on 31st of March 2021 with a VerisTechnologies U3 Soil Scanner (Salina, KS 67401, USA).

U3 was direct-contact soil scanner with EC (Electro Conductance), SOM (soil organic matter) and pH (soil acidity) for top soil (0–60 cm, 0–8 cm, and 0–8 cm, respectively). EC measurement was performed by direct electric-induced current conductivity measurement between two pairs of disc blades immersed in the ground. Immersion depth does not influence measurement but needed to be deep enough to secure reliable electric contact of the blades with measured soil. SOM measurement was provided by an optical sensor working in red and near-infrared wavelengths. The head of sensor moved in a shallow (5–8 cm) trench prepared securely with a preparation module installed in front of the scanner. The same trench was used by a pH module equipped with two electrodes and a dedicated hydraulic lift system. All U3 sensors were installed on the main frame providing the appropriate level and depth of work. Pulled by CanAm Traxter HD 10 Pro UTV (utility terrain vehicle, Bombardier Recreational Products Inc., Saint-Joseph Valcourt, Quebec, Canada), the U3 scanner maps fields with 15-m passages at a constant speed of 8–10 km. A dedicated application controls measurements quality and collects soil data. To ensure the highest spatial accuracy of the data, RTK GNSS (Real-time kinematic positioning using global navigation satellite system) receiver was used for positioning. All raw data, after automatic checking and cleaning, were calibrated in the Veris Technologies FieldFusion (Salina, KS 67401, USA) web application.

Soil cores for the composite soil sample were collected at zigzagging transects, which represented the areas of each zone. For each zone, about 15 soil cores were collected from the topsoil layer (0–30 cm in depth) and then mixed before the laboratory analyses. Standard laboratory analyses were performed for the evaluation of availability of plant (exchangeable) forms of phosphorus, potassium, magnesium, and soil pH. The measurement of pH was conducted by potentiometric method in KCl solution [27], the Egner–Riehm method was used for evaluation of potassium and phosphorus [28], and magnesium was determined using Schachtschabel method [29].

*2.3. Satellite Data*

In 2021, cloudless Sentinel-2 (Level-2A: bottom-of-atmosphere reflectances) multispectral satellite imagery from the beginning of May to mid-October were used for the analyses. The following dates were selected: 9th of May 2021, 4th of September 2021, 9th of September 2021, 4th of October 2021, and 11th of October 2021. The first date (9th of May 2021) was with bare soil, before the emergence of maize plants and the other dates were at intensive growth of maize (September) or at the end of vegetation period (October).

Three spectral indices were used for the analyses:

- Normalized difference vegetation index: NDVI = (NIR − RED)/(NIR + RED).
- Normalized difference water index 1: NDWI1 = (NIR − SWIR1)/(NIR + SWIR1).
- Normalized difference water index 2: NDWI2 = (NIR − SWIR1)/(NIR + SWIR1).

where NIR is near-infrared reflectance at central wavelength 833 nm; RED is reflectance at central wavelength 665 nm; SWIR1 is shortwave infrared reflectance at central wavelength 1610 nm; and SWIR2 is shortwave infrared reflectance at central wavelength 2190 nm. The NIR and RED bands had a spatial resolution of 10 m, and both SWIR bands had a spatial resolution of 20 m [30]. A 10-m buffer for each management zone was excluded from the analyses to remove the pixels of satellite images that overlapped with the borders of the fields. The mean values of the spectral indices were calculated using the zonal statistics tool in the QGIS software.

*2.4. Statistical Analysis*

Basic statistical parameters (means, ranges and standard deviations) were calculated for the variables that were analyzed in the study. Pearson's correlation coefficients were calculated between spectral indices (NDVI, NDWI1, and NDWI2) and variables acquired by soil proximal sensing (RED/IR ratio, EC, OM). The relationships between NDVI, nutrient content, and pH, as well between soil EC, nutrient content, and pH were evaluated using linear regression. The results were presented graphically together with the regression equations and the coefficients of determination ($R^2$). Multiple linear regression was applied for evaluation of the simultaneous effect of NDVI and soil EC on nutrient content and pH. The results were presented graphically with regression equations, $R^2$ and *p*-values. Statistical analyses were preformed using Statistica 13.3 program [31]. Significance level for all the analyses was set at 0.05 probability level.

## 3. Results

The soil reaction (pH) and the nutrient contents in soil were, in most parts of the fields, at high or very high levels (Table 1 and Figure 1), as recommended by crop-management advisory services [32]. The only exception was the content of magnesium, which was low or very low. The spatial pattern of the soil pH and the phosphorus, potassium, and magnesium contents were quite similar. These soil properties were moderately positively correlated (Table 2), and the high-pH zones tended to be high in all three nutrients (Figure 2). However, the correlation was not very strong (in the range from 0.12 between pH and K to 0.56 between K and Mg).

**Table 1.** Basic statistical parameters for variables used in the study.

| | Mean | Min. | Max. | SD |
|---|---|---|---|---|
| pH | 6.211 | 5.100 | 7.600 | 0.584 |
| Phosphorus (mg $P_2O_5$/100 g) | 16.294 | 8.200 | 35.000 | 5.985 |
| Potassium ($K_2O$ mg/100 g) | 24.021 | 15.700 | 35.000 | 5.585 |
| Magnesium (Mg mg/100 g) | 2.530 | 1.000 | 4.600 | 0.785 |
| NDVI 09/05/2021 | 0.143 | 0.116 | 0.204 | 0.022 |
| NDVI 04/09/2021 | 0.865 | 0.758 | 0.889 | 0.026 |
| NDVI 09/09/2021 | 0.845 | 0.742 | 0.869 | 0.031 |
| NDVI 04/10/2021 | 0.715 | 0.582 | 0.823 | 0.066 |
| NDVI 11/10/2021 | 0.536 | 0.432 | 0.674 | 0.070 |
| NDWI1 09/05/2021 | −0.155 | −0.193 | −0.060 | 0.027 |
| NDWI1 04/09/2021 | 0.447 | 0.329 | 0.488 | 0.030 |
| NDWI1 09/09/2021 | 0.410 | 0.290 | 0.452 | 0.035 |
| NDWI1 04/10/2021 | 0.244 | 0.121 | 0.389 | 0.068 |
| NDWI1 11/10/2021 | 0.012 | −0.047 | 0.112 | 0.044 |
| NDWI2 09/05/2021 | −0.158 | −0.206 | −0.045 | 0.036 |
| NDWI2 04/09/2021 | 0.699 | 0.570 | 0.743 | 0.034 |
| NDWI2 09/09/2021 | 0.667 | 0.491 | 0.720 | 0.047 |
| NDWI2 04/10/2021 | 0.498 | 0.328 | 0.649 | 0.076 |
| NDWI2 11/10/2021 | 0.246 | 0.159 | 0.373 | 0.057 |
| Red/IR ratio | 0.340 | 0.331 | 0.362 | 0.006 |
| EC (mS/m) | 8.091 | 1.813 | 16.332 | 3.458 |
| OM | 1.388 | 1.259 | 1.617 | 0.069 |

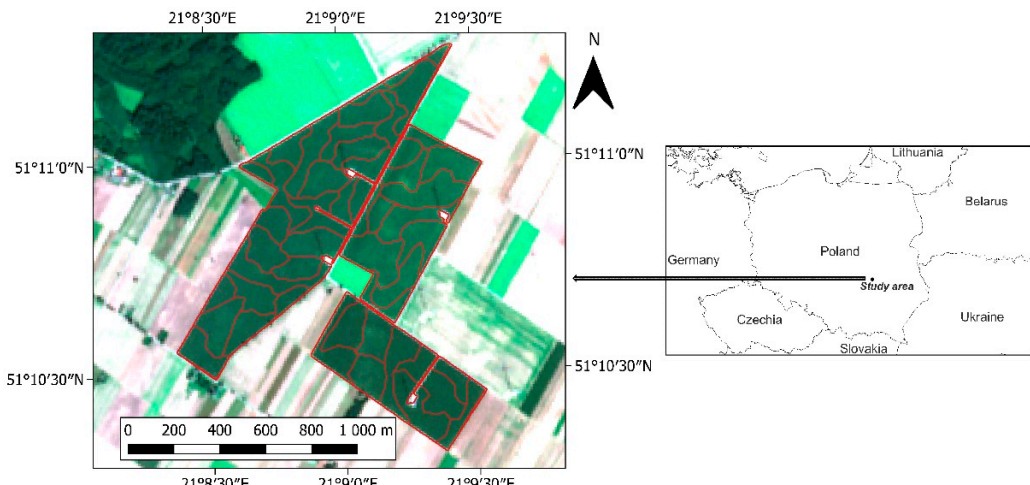

**Figure 1.** The studied fields with delineated management zones and their locations.

**Table 2.** Correlation coefficients between nutrient content, pH and spectral indices, EC, and organic-matter content (significant correlations at 0.05 probability level are marked in red).

|  | pH | Phosphorus (mg/100 g) | Potassium (mg/100 g) | Magnesium (mg/100 g) |
|---|---|---|---|---|
| NDVI 09/05/2021 | 0.379 | 0.610 | 0.581 | 0.453 |
| NDVI 04/09/2021 | −0.175 | −0.359 | −0.192 | −0.117 |
| NDVI 09/09/2021 | −0.170 | −0.340 | −0.195 | −0.120 |
| NDVI 04/10/2021 | −0.170 | −0.282 | 0.270 | −0.136 |
| NDVI 11/10/2021 | −0.141 | −0.233 | 0.366 | −0.150 |
| NDWI1 09/05/2021 | 0.228 | 0.363 | 0.339 | 0.000 |
| NDWI1 04/09/2021 | −0.103 | −0.283 | −0.248 | 0.145 |
| NDWI1 09/09/2021 | −0.123 | −0.283 | −0.259 | 0.089 |
| NDWI1 04/10/2021 | −0.204 | −0.355 | 0.227 | −0.234 |
| NDWI1 11/10/2021 | −0.232 | −0.348 | 0.259 | −0.303 |
| NDWI2 09/05/2021 | 0.202 | 0.355 | 0.443 | 0.070 |
| NDWI2 04/09/2021 | −0.140 | −0.280 | −0.326 | 0.103 |
| NDWI2 09/09/2021 | −0.126 | −0.231 | −0.304 | 0.088 |
| NDWI2 04/10/2021 | −0.258 | −0.370 | 0.192 | −0.200 |
| NDWI2 11/10/2021 | −0.305 | −0.393 | 0.232 | −0.291 |
| Red/IR ratio | −0.080 | 0.178 | 0.565 | 0.195 |
| EC (mS/m) | 0.000 | 0.167 | 0.801 | 0.479 |
| SOM | −0.152 | 0.214 | 0.656 | 0.352 |
| pH |  | 0.460 | 0.123 | 0.379 |
| Phosphorus (mg/100 g) | 0.460 |  | 0.319 | 0.432 |
| Potassium (mg/100 g) | 0.123 | 0.319 |  | 0.562 |
| Magnesium (mg/100 g) | 0.379 | 0.432 | 0.562 |  |

The means, ranges, and standard deviations (SD) for the three spectral indices (NDVI, NDWI1, and NDWI2) were evaluated for five terms in the study area. In the first term (09/05/2021), the means of all the spectral indices were the lowest, because it took place when the soil was bare after sowing but before the emergence of plants. During the period from June to August, cloudy conditions occurred, and there was no cloudless satellite imagery available for analysis. Two satellite images from the beginning of September were used; this was the period of the early ripening of maize (81–83 stage according to the BBCH growth scale). The values of the spectral indices for this stage were very high (with a mean NDVI of about 0.85–0.86), which indicates very intensive crop growth and a high green biomass per unit area. The two subsequent terms for the evaluation of the spectral indices were in the beginning of October, i.e., the senescence stage of the maize plants (91–95 stage

according to the BBCH growth scale), when the plants were only partially green and the spectral indices had lower values.

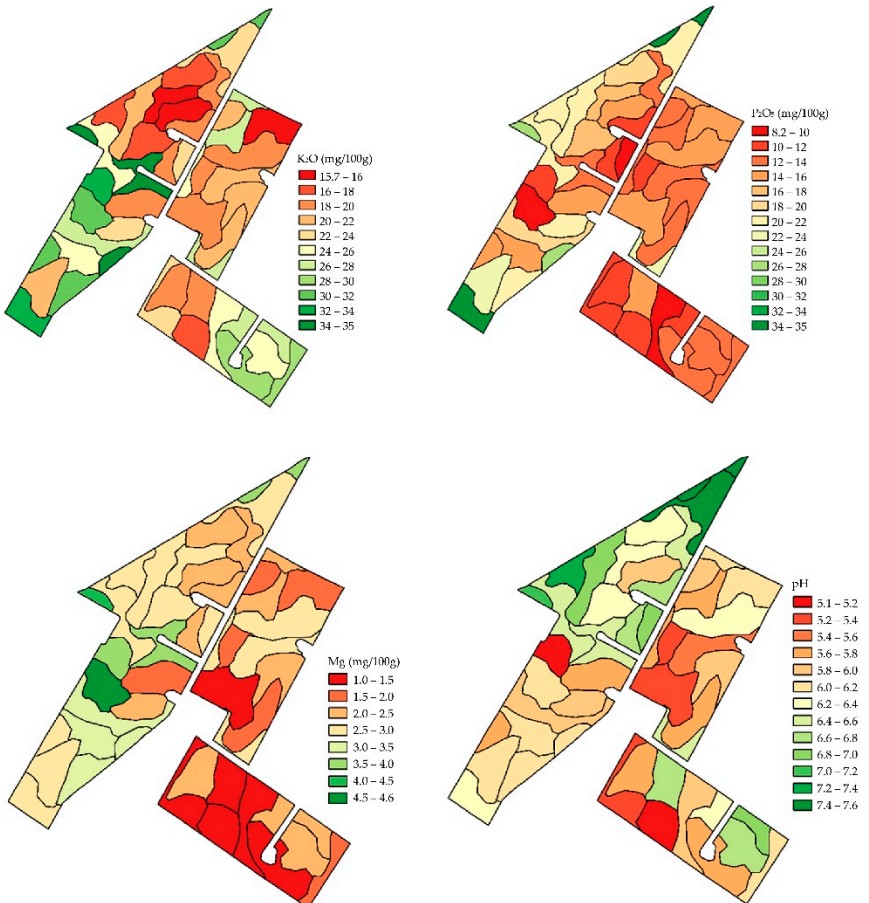

**Figure 2.** Nutrient content in soil and pH in study area.

Three variables obtained by soil proximal sensing were studied, i.e., the apparent electrical conductivity of the soil, the soil organic-matter content, and the ratio of the soil reflectance in red and near-infrared wavelengths (Red/IR ratio) (Figure 3). The variable with the highest spatial variability was the EC, characterized by a high standard deviation in relation to the mean. The spatial variability of the soil organic matter and the red/IR ratio were much lower.

The results of the correlation analysis presented in Table 2 prove statistically significant, strong, or moderate positive relationships between the soil pH, nutrient content, and NDVI of the bare soil (NDVI in 9th of May 2021). The relationships between the NDVI and pH, along with the NDVI and nutrient content, are presented graphically together with regression functions in Figure 4. The strongest relationships were observed between the NDVI and phosphorus (r = 0.61), as well as the NDVI and potassium (r = 0.58).

The relationships between the NDVI, evaluated in terms of later plant development, and the pH and nutrient content were much weaker, usually insignificant. The correlations between the pH, as well as the nutrient content, and the other two spectral indices, i.e., NDWI1 and NDWI2, were relatively weak and inconclusive; sometimes, they were positive, and other times, they were negative.

The relationships between the variables obtained by soil proximal sensing (EC, red/IR ratio and SOM) and the potassium and magnesium content were positive, and almost all were statistically significant (Table 2, Figure 5). The strongest correlation was observed between the EC and the potassium content (r = 0.80). Slightly weaker, but still strong, were the correlations between the red/IR ratio and potassium (r = 0.56), along with the

correlations between SOM and potassium (r = 0.66). The correlation between the EC and magnesium was moderate (r = 0.48), and the correlation between SOM and magnesium was slightly weaker, but significant (r = 0.35). The correlations between the red/IR ratio, EC, SOM, pH, and phosphorus were weak and insignificant.

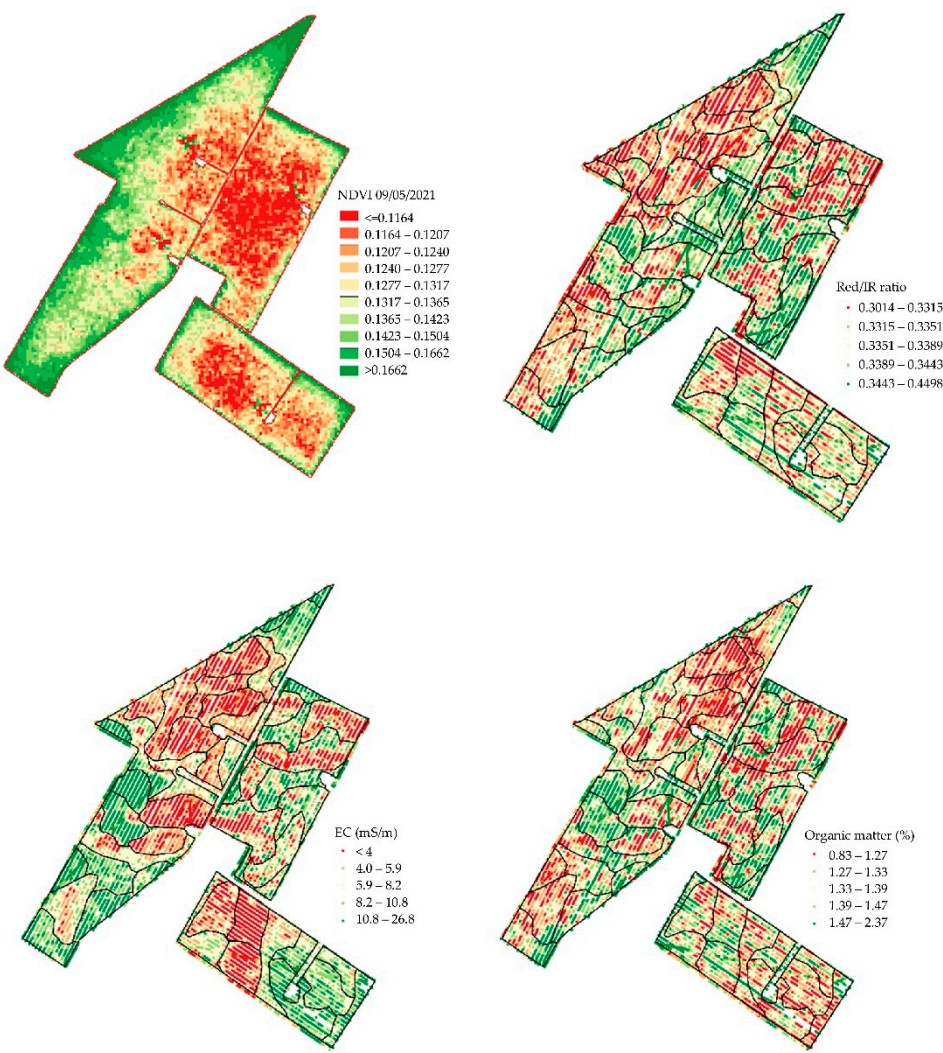

**Figure 3.** NDVI of bare soil and maps based on soil scanning using Veris ((Salina, KS 67401, USA) machinery (EC, red/IR ratio and estimated organic-matter content).

In addition to the univariate analyzes, a multiple regression analysis was performed to assess the simultaneous effect of the EC and NDVI (on 09/05/2021) on the soil pH and nutrient content. The relationships are presented graphically in Figure 6. The strongest relationship was observed for potassium, for which the simultaneous effect of the EC and NDVI explained 66% of the total variability ($R^2$ = 0.663 for multiple regression). A weaker, but quite strong, relationship was observed between the EC and NDVI for the content of phosphorus ($R^2$ = 0.425). The relationships between the EC and NDVI with contents of potassium and soil pH were weaker, but still significant ($R^2$ 0.280 and 0.216, respectively).

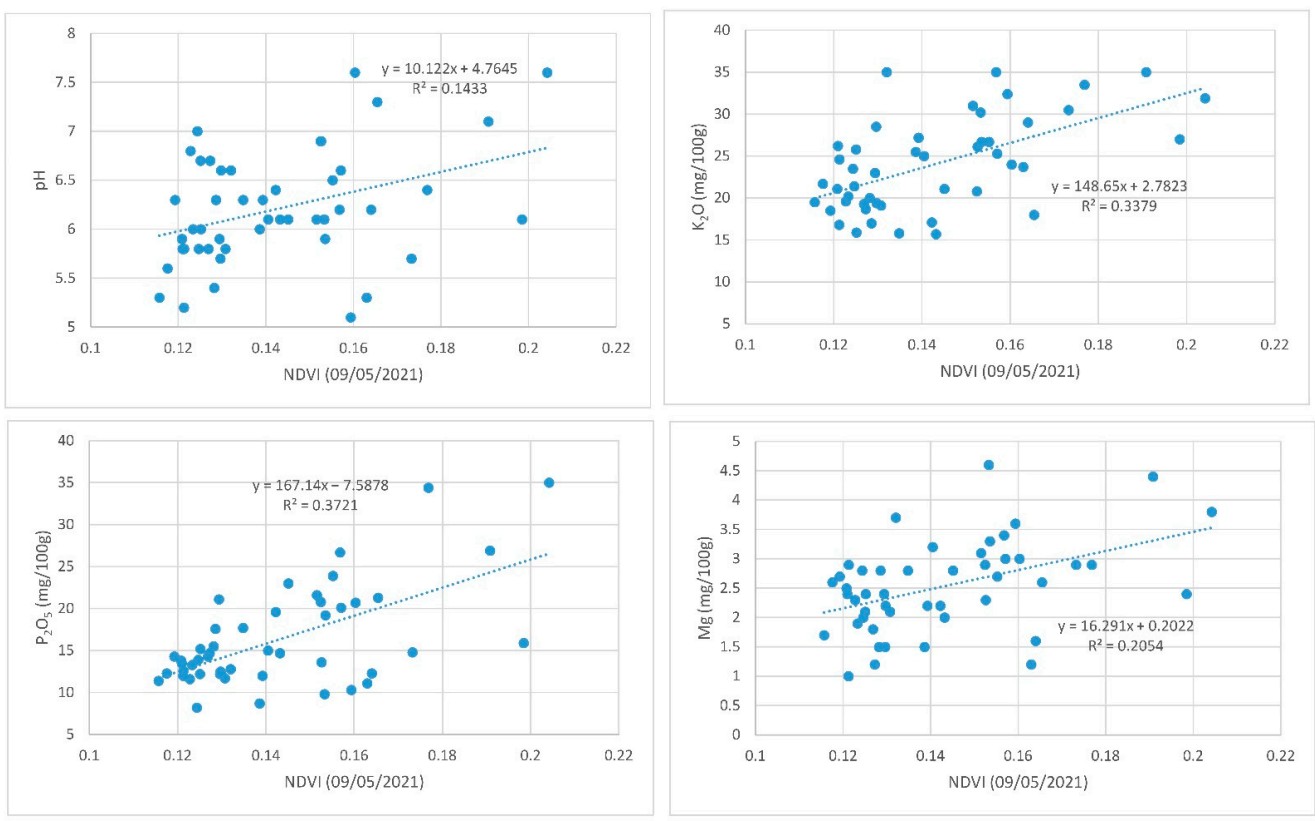

**Figure 4.** The relationships between NDVI and pH, as well as nutrient content in soil evaluated using simple linear regression.

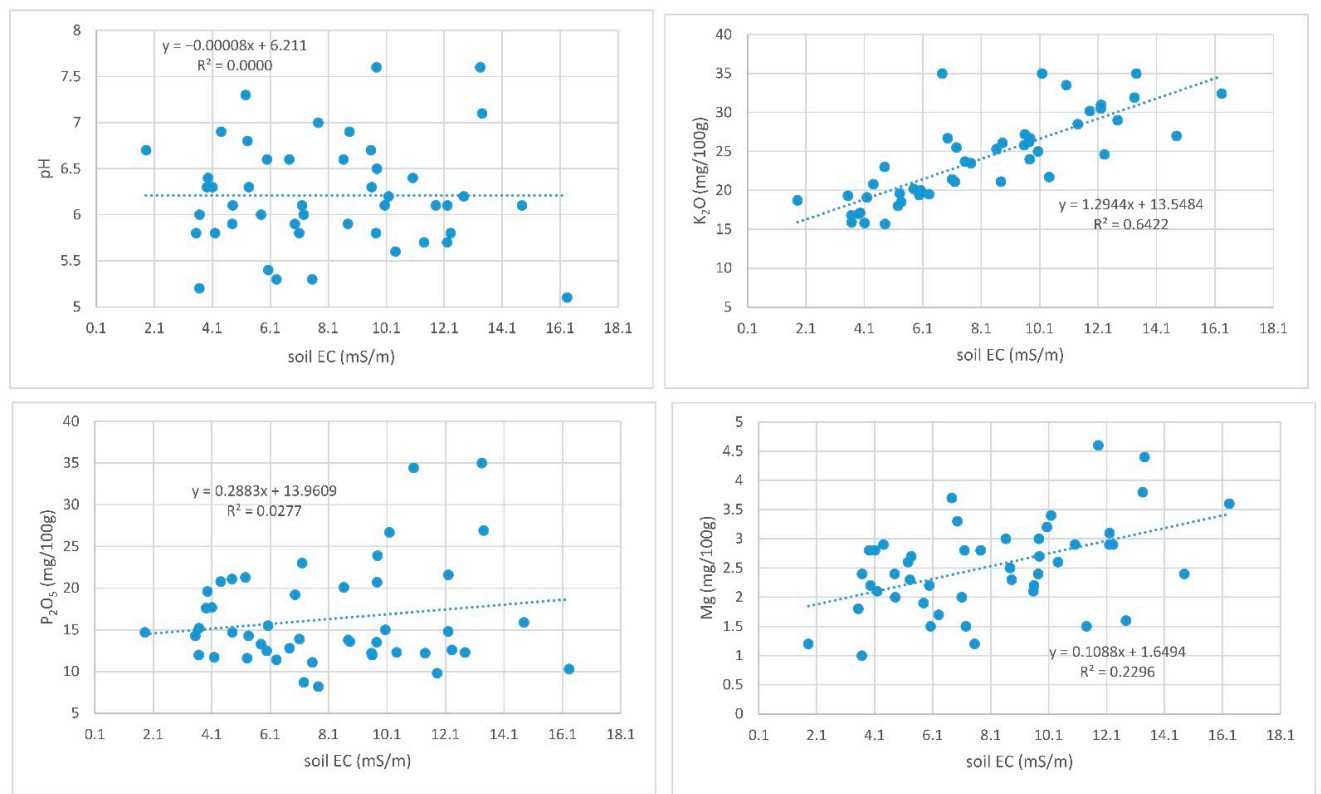

**Figure 5.** The relationship between soil EC and pH, as well as nutrient content, in soil evaluated using simple linear regression.

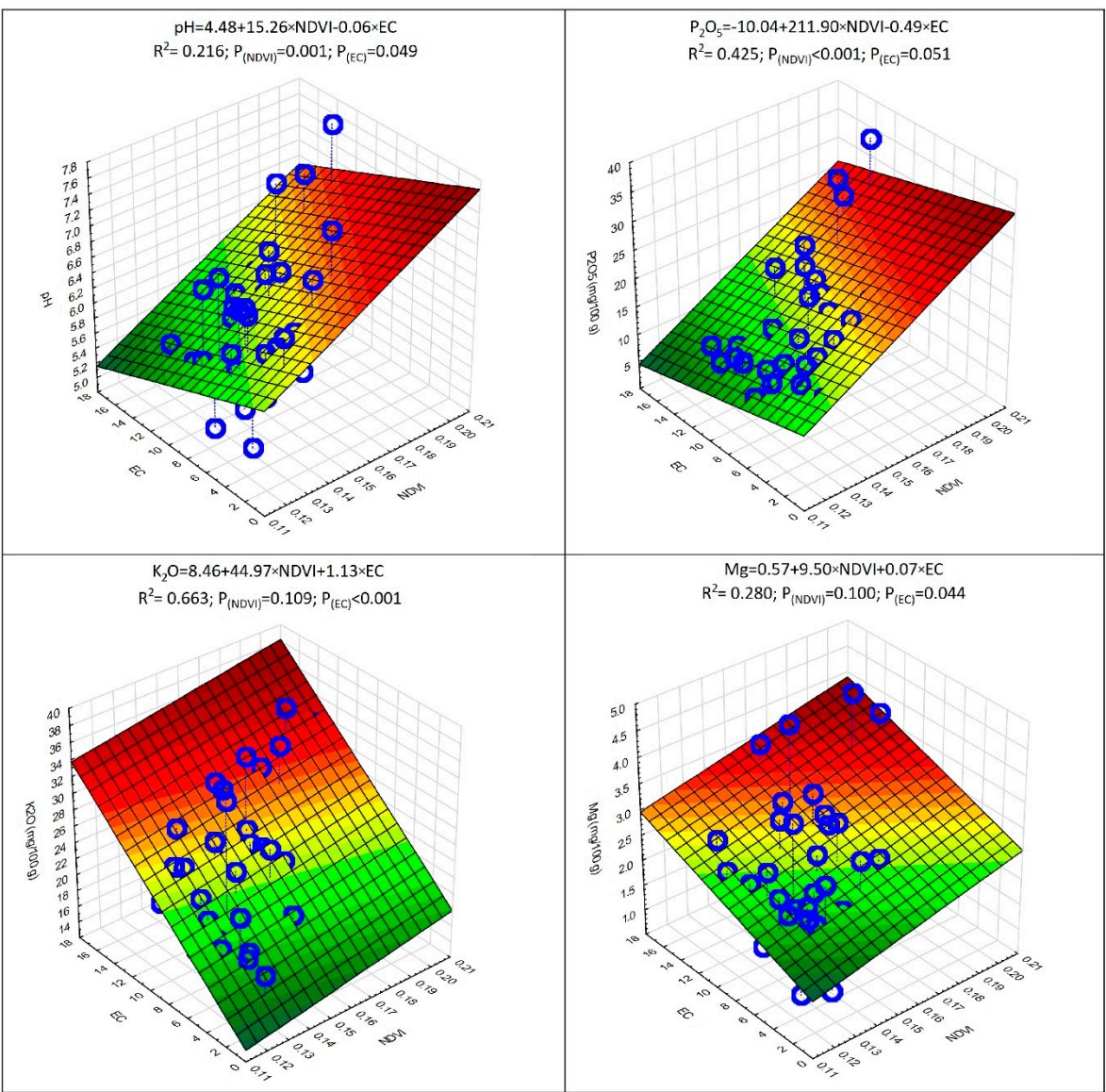

**Figure 6.** The relationship between soil EC and NDVI (at 09/05/2021) and pH and nutrient content in soil evaluated using multiple linear regression.

## 4. Discussion

The results obtained in this study proved the existence of significant positive relationships between the NDVI and pH, as well as between the NDVI and the nutrient content in the soil, a very strong positive correlation between the EC and potassium, and a moderately positive correlation between the EC and magnesium. The NDVI, for which significant relationships were proven, was evaluated for bare soil (after the sowing of maize, but before its emergence). The correlations ranged from 0.38 (NDVI with pH) to 0.61 (NDVI with phosphorus). Most of the correlations evaluated between the NDVI during the vegetation period of the maize and the nutrient content and pH were weak and insignificant. The correlations between the NDWI (both NDWI1 and NDWI2) and the nutrient content, as well as between the NDWI and pH, were inconsistent, i.e., most were not significant and some were significant, while some were positive and others were negative.

Significant, positive relationships between the Sentinel-2 derived NDVI and the potassium content in the soil, as well as between the EC and potassium, were obtained in our previous study [33], where the NDVI during vegetation cover (winter cereals or winter rapeseed) was used for the analyses. In a study conducted with spring wheat [34], a

significantly strong positive correlation (r = 0.61) was obtained between soil-exchangeable potassium and the Landsat-8-derived NDVI. Moreover, in this study, a very strong correlation (r = 0.92) between the NDVI and grain yield was observed, which means that the potassium content was strongly correlated with both the NDVI and the grain yield of the spring wheat. Furthermore, in the same study, the correlation between the NDVI and easily available phosphorus was much weaker and not significant (r = 0.35). Positive significant relationships between the NDVI measured by the Crop Circle sensor for wheat during the booting and heading growth stages and the potassium content in the soil were observed by Whetton et al. [35]. In the same study, positive significant relationships between the NDVI and the content of potassium in soil were observed for barley at the booting and heading stages. The correlation between the available potassium in soil and the NDVI is not always positive. In a study conducted on a large spatial scale [36] (in which the distance between the most faraway sampling points was about 30 km), the correlation between the satellite (MODIS)-derived NDVI and the potassium available in the soil was negative and very weak (r = −0.08), but still significant, because of the large sample size.

A significant positive correlation between the soil EC and both phosphorus and potassium was found in the study by Omonode and Vyn [17]. The correlation coefficients were in a range from 0.16 to 0.60, depending on the depth of the soil layer and year. A significant positive correlation (r = 0.42) between the EC and the potassium content in soil was obtained by Li et al. [37]. In the same study, a weak but negative correlation between EC and phosphorus content was found. The correlations were similar to these obtained in our previous study, where a stronger positive correlation was found between the EC and potassium and a weaker negative correlation was observed between the EC and phosphorus.

The correlation coefficients between the EC and the content of potassium and phosphorus are not always unequivocal. In the study by Heiniger et al. [38], the correlation coefficients between the EC and potassium were in a very wide range, from −0.75 to 0.70, and those between the EC and phosphorus were from −0.61 to 0.70. In the study by Tarr et al. [39], the correlation coefficients between the EC and potassium, as well as between the EC and phosphorus, were negative: r = −0.56 and r = −0.06, respectively. In the same study, a positive correlation was found between the EC and pH (r = 0.70).

In this study, the analysis of multiple linear regression proved that the regression model, in which the EC and NDVI for the bare soil were predictors of pH and nutrient content, made it possible to obtain stronger relationships, especially for the content of potassium in soil ($R^2$ = 0.663) and phosphorus ($R^2$ = 0.425), used as dependent variables. This means that the prediction of the within-field variability of potassium and phosphorus can be better if EC and NDVI are used as predictors of phosphorus and potassium or as variables for the delineation of homogenous within-field management zones for soil sampling.

The usefulness of both the NDVI and soil EC was proven in the studies by Esteves et al. [21,22], in which the management zones delineated based on the NDVI and soil EC differed significantly, depending on the most important agronomic soil properties, including the soil texture, soil pH, extractable phosphorus, and potassium. This confirms that the delineation of management zones based on the NDVI and soil EC can be used for the variable-rate application of fertilizers, as well as for optimized soil sampling. Another study that proves the usefulness of NDVI and soil EC for the delineation of within-field homogenous zones for variable application rates and optimized soil sampling is the study conducted in pastures by Serrano et al. [26]. In this study, a very strong positive correlation was observed between the NDVI and NDWI, which was not detected in our study. Many other studies proved the usefulness of the NDVI and soil EC in the multivariate delineation of management zones [23–25]. The NDVI and EC in these studies were derived using various methods, i.e., NDVI measured by satellite or UAV sensors, soil EC measured with soil sensors in direct contact with the soil or by electromagnetic induction. All these types of data have similar usefulness.

The delineation of within-field management zones can be based on various soil properties or yield-related traits such as the EC, pH, nutrient content (especially N, P, and K), soil organic matter, vegetation indices, and crop yield [26,40,41]. Limiting the number of variables used for the delineation of management zones is important in agronomical practice. Simplified approaches can be used when only one variable, such as the NDVI or crop yield, is available for the classification of similar areas within a field [42,43]. A common approach in management-zone delineation is the use of two variables characterizing soil properties (EC) and crop condition (yield) [2]. Because crop yield is usually very strongly correlated with vegetation indices, such as the NDVI [44], crop yield can be replaced by the spectral index in the process of management-zone delineation.

### 5. Conclusions

Significant positive correlations were observed between the NDVI for bare soil and nutrient content (phosphorus, potassium, and magnesium) in soil and pH. A very strong positive correlation was observed between the soil EC and the potassium content and a moderate correlation was noted between the soil EC and the magnesium content. The multiple-regression analysis proved that the soil nutrient content, especially of potassium and phosphorus, was strongly related to the EC and NDVI. We suggest the use of these two variables for the delineation of management zones in agronomic practice for optimized soil sampling and variable-rate fertilization, especially in the case of potassium and phosphorus fertilization.

**Author Contributions:** Conceptualization, D.G. and P.M.; methodology, P.M.; validation, P.M., D.G. and E.W.-G.; formal analysis, P.M.; investigation, P.M.; resources, P.M.; data curation, P.M.; writing—original draft preparation, P.M. and D.G.; writing—review and editing, E.W.-G.; visualization, P.M.; supervision, D.G.; All authors have read and agreed to the published version of the manuscript.

**Funding:** This research received no external funding.

**Institutional Review Board Statement:** Not applicable.

**Informed Consent Statement:** Not applicable.

**Data Availability Statement:** The data presented in this study are available on request.

**Conflicts of Interest:** The authors declare no conflict of interest.

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
