# Peer review of "Soil Electrical Conductivity and Satellite-Derived Vegetation Indices for Evaluation of Phosphorus, Potassium and Magnesium Content, pH, and Delineation of Within-Field Management Zones"

_agriculture, doi:10.3390/agriculture12060883_

Round 1

Reviewer 1 Report

Dear authors,

The manuscript evaluated "Soil Electrical Conductivity and Satellite Derived Vegetation Indices for Evaluation of Phosphorus, Potassium and Magnesium Content, pH and Delineation of Within-Field Management Zones”. The topic is relevant, and the authors did a lot of work, and the methodology used is adequate to the objective of the study. However, some revisions are still required as shown below.

Abstract

The novelty of this study and a solid conclusion regarding the obtained results should be given.

Introduction

The significance of the study and a solid hypothesis of the present study at the end of the introduction should be provided to give the reader more information regarding the mechanistic used to achieve the objective of the study.

The introduction should also include the recent literature discussing the research hypothesis. Please include the related recent literature.

Methods

The methods should be written in more detail to be reproducible.

Discussion

The discussion should be interpreted with the results as well as discussed in relation to the present literature.

The references section should include studies about the recent literature presented on this topic as well.

Reviewer 2 Report

I have evaluated the work by Mazur who attempted to establish the relationships between selected spectral indices and analyzed soil physicochemical data. Generally, the work is well structured, appropriate methods were used and results are presented nicely.

Minor corrections required:

1. All abbreviations should be defined. Example: UAV, RTK GNSS, etc.

2. Names and place of manufacturers of special equipment used should mentioned. Example: VerisTechnologies U3 Soil 85 Scanner (line 85), and others. 

Reviewer 3 Report

The results were very interesting and corroborated with the literature. However, the author (s) left something to be desired in the discussion of the results. They could have explored the discussion better, emphasizing the importance of this study, for example.

Round 2

Reviewer 1 Report

Dear authors,

The manuscript revised and improved according to suggestions. I recommend acceptance.

With kind regards!